# Towards colloidal spintronics through Rashba spin-orbit interaction in lead sulphide nanosheets

Mohammad Mehdi Ramin Moayed[1], Thomas Bielewicz[1], Martin Sebastian Zöllner[2], Carmen Herrmann[2] & Christian Klinke[1]

Employing the spin degree of freedom of charge carriers offers the possibility to extend the functionality of conventional electronic devices, while colloidal chemistry can be used to synthesize inexpensive and tunable nanomaterials. Here, in order to benefit from both concepts, we investigate Rashba spin–orbit interaction in colloidal lead sulphide nanosheets by electrical measurements on the circular photo-galvanic effect. Lead sulphide nanosheets possess rock salt crystal structure, which is centrosymmetric. The symmetry can be broken by quantum confinement, asymmetric vertical interfaces and a gate electric field leading to Rashba-type band splitting in momentum space at the M points, which results in an unconventional selection mechanism for the excitation of the carriers. The effect, which is supported by simulations of the band structure using density functional theory, can be tuned by the gate electric field and by the thickness of the sheets. Spin-related electrical transport phenomena in colloidal materials open a promising pathway towards future inexpensive spintronic devices.

[1] Institute of Physical Chemistry, University of Hamburg, 20146 Hamburg, Germany. [2] Institute of Inorganic and Applied Chemistry, University of Hamburg, 20146 Hamburg, Germany. Correspondence and requests for materials should be addressed to C.K. (email: klinke@chemie.uni-hamburg.de).

Though in the last decades developments in semiconductor device engineering led to constantly improved performances, nowadays it is very difficult to keep this pace[1,2]. Further challenges in terms of manufacturing cost, flexibility, power consumption and device functionality are ahead, for example, for the internet of things[2,3]. Recent ideas try to address these issues by the introduction of new materials as the active semiconductor channel or new mechanisms of information processing[2,4]. One solution is replacing the electron charge by its spin as an information carrier[4,5]. For this purpose, materials with strong spin–orbit coupling (SOC) are particularly interesting since in these materials spin orientation can be manipulated by an external electrical field[5–8]. In semiconductors, SOC can originate from bulk inversion asymmetry (Dresselhaus SOC) or from structural inversion asymmetry (Rashba SOC)[5–7,9,10]. Since Rashba SOC can be influenced by customizing the structure (for example, by confining the material) or by a gate voltage, it is being studied comprehensively[5–7,10]. As a further development in data processing, the concept of valleytronics has been introduced for materials with multiple valleys (multiple extrema with equal energies) in the band structure[11–14]. By controlling the number of carriers in each valley, a so-called valley-polarized current can be produced, which leads to an additional degree of freedom[4,12–14]. As a consequence of spin–valley coupling in these materials, it has been shown that the valleys can be also populated selectively based on the spin of the carriers[11–13,15].

Up to now, spin/valley-dependent transport phenomena have been investigated mostly in two-dimensional nanostructures prepared by industry-incompatible methods (for example, mechanical exfoliation) or by cost-intensive instruments such as molecular-beam epitaxy[6,9,16]. In this work, we demonstrate the presence of easily accessible SOC in nanomaterials synthesized by colloidal chemistry. The solution-based synthesis of colloidal nanomaterials offers enormous opportunities in the production of inexpensive and high-quality crystals for electronic and, as we will show, spintronic devices[17–19]. Colloidal lead sulphide (PbS) nanosheets as continuous 2D crystals with promising properties do not suffer from tunnel barriers like other colloidal systems, such as thin films of nanoparticles[17,18,20]. However, their rock salt crystal structure is centrosymmetric. In order to break the inversion symmetry, we apply an electric field (gate) to the crystal as well as different boundaries underneath and above the PbS nanosheet (SiO$_2$ and vacuum)[6,21]. In combination with strong SOC, this suggests Rashba-type band splitting, which is confirmed by Kohn-Sham density functional theory (KS-DFT). In PbS nanosheets, the band splitting occurs in momentum space at the four M points. Upon illumination with circularly polarized light, the circular photo-galvanic effect (CPGE) leads to a net current that can be explained by an unconventional selection mechanism, depending on the spin orientation of the carriers. The observed effect is precisely tunable by changing the gate voltage or by changing the confinement (thickness of the sheets).

## Results

**Circular photo-galvanic measurements on the nanosheets**. We synthesized PbS nanosheets with lateral dimensions of up to 5 μm and tunable thickness of 2–20 nm by introducing different amounts of oleic acid to the synthesis[19,20,22] (for crystallographic characterization see Supplementary Fig. 1). The height of the

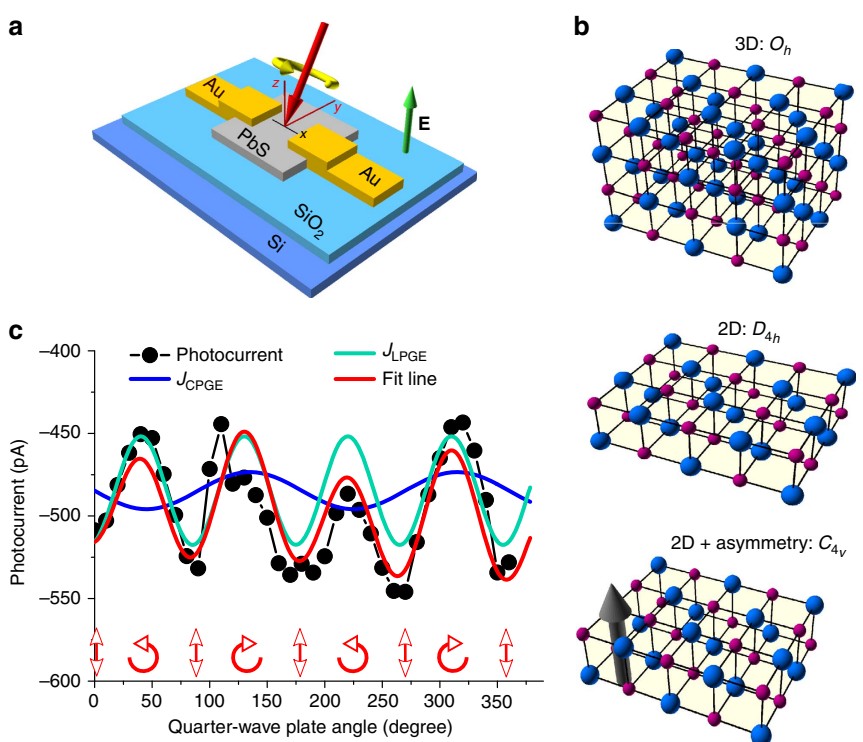

**Figure 1 | Circular photo-galvanic effect in lead sulfide nanosheet devices.** (**a**) Schematic image of the experimental setup. The semiconductor channel (PbS nanosheet, shown in grey) experiences an electric field and asymmetric interfaces (vacuum and SiO$_2$, shown in blue) and is illuminated with circularly polarized light. (**b**) Breaking the inversion symmetry in a PbS crystal. The symmetry is reduced from $O_h$ point group (for the bulk crystal) to $C_{4v}$ (2D crystal with external asymmetries). The blue and violet spheres represent lead and sulfur atoms respectively. The arrow shows the asymmetry in the crystal, including the gate electric field and the asymmetric interfaces. (**c**) Photocurrent at zero bias as a function of the quarter-wave plate angle (with a small non-zero incidence illumination angle). Fitting the results shows the existence of a non-zero CPGE current, suggested to originate from Rashba SOC in PbS nanosheets (LPGE: linear photo-galvanic effect). The arrows show the quarter-wave plate angles, in which the light is circularly or linearly polarized.

nanosheets directly determines the band gap of the material by confinement[23]. This tunability is of advantage for the performance of applications such as transistors, photodetectors and solar cells[17,18,20,23]. The synthesis of the nanosheets is independent of the fabrication of the devices, which makes it practical for industrial applications. The nanosheets were spin-coated on Si/SiO$_2$ substrates and contacted individually with Au electrodes by electron-beam lithography (see Supplementary Fig. 2). The contacts were placed in <110> direction of the crystal (corresponding to the M point direction in **k**-space) which is also the lateral growth direction of the nanosheets.

In order to evidence the possibility of spin manipulation in PbS nanosheets, we performed measurements on the CPGE as a function of light polarization[6,7,9–12]. For this purpose, photo-excited charge carriers with preferred spin orientation were prepared by illumination with circularly polarized light produced by a linearly polarized laser beam ($\lambda = 627$ nm) through a quarter-wave plate. The helicity of the exciting light was determined by the angle of the quarter-wave plate, which controls the spin orientation of the carriers. The beam was pointed to the sample obliquely in the $yz$ plane ($x$ is the direction of the current flow, $z$ is normal to the nanosheet and $y$ is perpendicular to these two).

Figure 1a schematically illustrates the experimental setup. This configuration provides the required inversion asymmetry (including the gate electric field and asymmetric vertical boundaries) in order to detect the CPGE. Bulk PbS with a rock salt crystal structure obeys the $O_h$ point group symmetry which is inversion symmetric. By confining the material in $z$ direction (height of the nanosheets), the symmetry is reduced first to the $D_{4h}$ point group and then, by application of asymmetric vertical interfaces on top and underneath (SiO$_2$ and vacuum) as well as by the gate electric field, to $C_{4v}$. For this symmetry group, the inversion centre is absent, which supports the band splitting by SOC (Fig. 1b). Figure 1c depicts the variation of the photocurrent with changing the angle of the quarter-wave plate. It shows that illumination of the unbiased devices can yield a non-zero helicity-dependent photocurrent whose direction can be reversed by changing the light polarization from right-handed to left-handed, which is a sign for a spin-polarized current, generated as a consequence of spin injection into a spin–orbit coupled system[5,6,12]. In addition to that, other polarization-independent currents are detected which should be distinguished from the spin-related current.

The generated photocurrent is described by the expression

$$J_{\text{total}} = J_0 + J_{\text{CPGE}}\sin(2\varphi) + J_{\text{LPGE}}\sin(2\varphi)\cos(2\varphi) \qquad (1)$$

which can be employed for fitting the measurement results to determine the contribution of each effect[5,9–12,16]. In the equation, $J_0$ is the background current, resulting from, for example, photovoltaic effects, the Damber effect or hot electron injection. This component is independent of the helicity of the light. $J_{\text{LPGE}}$ is the amplitude of the linear photo-galvanic effect which is due to asymmetric scattering of electrons along the conduction path. Although the LPGE is a function of the quarter-wave plate angle, it is equal for the right-handed and left-handed polarized light, which implies its independency from the helicity or respectively the spin orientation of the carriers. The oscillation period of the LPGE is equal to 90°. Eventually, $J_{\text{CPGE}}$ is the amplitude of the CPGE, the current which is attributed to the population of the conduction band with spin-polarized charge carriers. In contrast to the LPGE, the CPGE oscillates with a period of 180°. It is maximum at a helicity of $+1$ (45° and 225°), minimum when the helicity is $-1$ (135° and 315°) and zero when the light is linearly polarized (0°, 90°, 180° and 270°) (refs 11,16).

**Origin of the effect.** Generally, different phenomena can lead to the existence of a CPGE in a 2D material, including Rashba/Dresselhaus SOC[5,6,9], orbital interactions[16], spin–valley coupling[11,12] or topological surface states[24]. Even in centrosymmetric materials with negligible SOC, in silicon or graphene for instance, this effect has been detected due to the asymmetry at the edges or contacts[16,25] (in-plane asymmetries). Lead sulphide as a crystal including high atomic-number Pb atoms ($Z(\text{Pb}) = 82$) can be considered as having strong SOC[15,26]. Further, the band structure of 2D PbS consists of a rectangular Brillouin zone with four equal valleys at the corners, the M points. By selectively exciting the carriers in each valley, a valley-polarized current might occur[11–13,15,27]. Such effects can be detected by the CPGE current since the angular momentum of

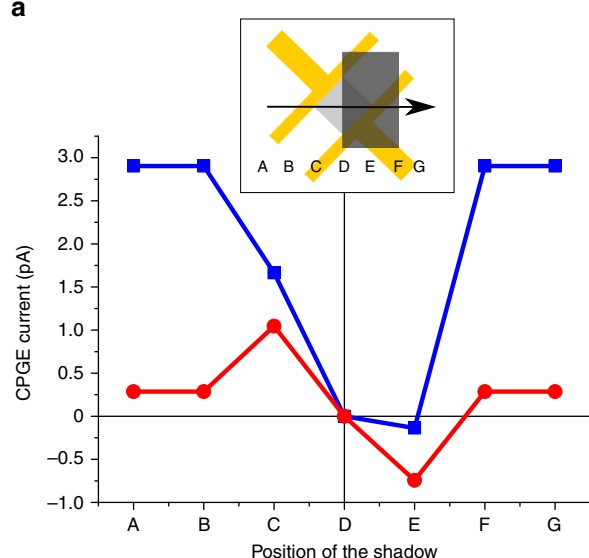

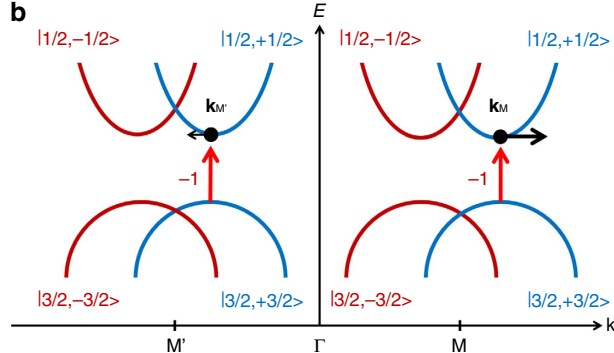

**Figure 2 | Origin of the circular photo-galvanic effect in lead sulfide nanosheets.** (**a**) The generated CPGE by shadowing different positions of the device, and with different illumination angles. The blue and the red lines represent the high and the low illumination angles respectively. The letters indicate the position of the centre of the shadow. The contribution of the vertical asymmetry and the enhanced in-plane asymmetry can be observed ($I_{\text{vertical}}$ and $I_{\text{in-plane}}$). The inset is a schematic top view of the device to illustrate the position of the shadow during the experiment. In position D the whole device is covered, in C and E half of the device ($I_{\text{shadow}}$), and A, B, F, and G represent the full illumination of the device ($I_{\text{full}}$). (**b**) Illustration of the possible selection mechanism for exciting the carriers over the band gap. Illumination of the PbS crystal with circularly polarized light leads to transitions in both valleys but excitation of only one spin orientation. ($\mathbf{k}_{\text{M,M'}}$: momentum of the excited carriers in a valley). Here, the angular momentum of exciting photons is $-1$, which is added to the spin-angular momentum of electrons. The states are labeled as $|J, M_J\rangle$.

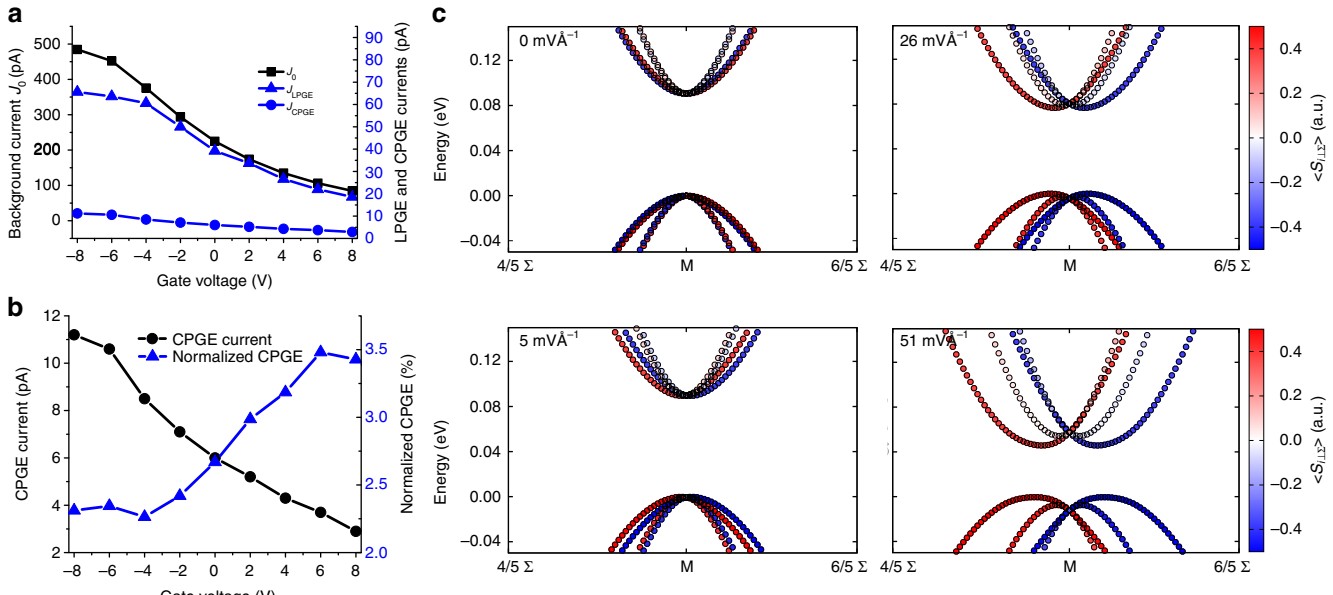

**Figure 3 | Dependency of the effect on the back-gate voltage.** (**a**) Dependence of different components of the photocurrent on the gate voltage. Similar gate dependencies can be observed for $J_0$, $J_{LPGE}$ and $J_{CPGE}$ (background current, linear photo-galvanic effect and circular photo-galvanic effect). (**b**) The measured CPGE current and its normalized value as a function of back-gate voltages for an 18 nm thick sheet. (**c**) Calculated (DFT) valence and conduction bands at the M point with application of different external electric fields—modelling the gate (15 layers). Σ denotes the path from Γ to M. The red and the blue colours denote the sign of the spin projection on the in-plane axis perpendicular to the Γ–M path ($<S_{i\perp\Sigma}>$).The valence-band maximum was set to 0 eV.

circularly polarized photons couples to the spin or valley index of the carriers, or leads to asymmetric scattering of the carriers. In any case, an inversion asymmetry must be present in the structure, which results in an asymmetric distribution of carriers in momentum space or asymmetric movement of the carriers in a certain direction. Therefore, a net CPGE current can be observed[5,6,9–12,16,25].

As the first step to explain the effect, we show that it mainly originates from the imposed vertical asymmetry to the crystal and not from the contacts/edges. For this purpose, we measured the CPGE current while we shadowed different parts of the device, to control the position of the beam and to intentionally enhance the asymmetry between the contacts/edges. Figure 2a shows the measured CPGE according to the position of the shadow. The measurements were done with a high incidence angle (about 15 degrees, the max. incidence angle due to the size of the view port of the vacuum chamber) and a lower one (around 6 degrees). By moving the shadow from one side of the device to the other one under the low incident angle, it is possible to reverse the direction of the in-plane asymmetry[12,16,24,25], expressed by a sign change of the shadowed CPGE current $I_{shadow}$ (Fig. 2a, positions C and E). On the other hand, the function shows an offset which implies that a part of the current is independent of the in-plane asymmetry. In the following, this part will be shown to be the current originating from the vertical asymmetry ($I_{vertical}$), caused by the two interfaces and the gate. By increasing the incidence angle, firstly it can be observed that the unshadowed current ($I_{full}$) increases, which shows that the vertical asymmetry is an effective factor for generating $I_{full}$. In fact, to observe the CPGE based on in-plane asymmetry ($I_{in-plane}$), illumination should be perpendicular to the surface of the nanosheets. For a vertical asymmetry, such illumination leads to a zero CPGE current, and $I_{vertical}$ increases with increasing the incidence angle[6,12,16,24–26], which explains the increase of the measured $I_{full}$ with the larger angle. When the shadow is applied, the imbalance of $I_{shadow}$ (difference between the currents in positions C and E) becomes

more pronounced with the higher illumination angle. It shows that $I_{vertical}$ has a larger part in $I_{shadow}$, while $I_{in-plane}$ is almost constant. By shadowing, we cover half of the device, which decreases the effective part of the crystal to generate $I_{vertical}$. In contrast, it leads to a higher $I_{in-plane}$, as we magnify the asymmetry. Since the share of $I_{vertical}$ is larger now, $I_{shadow}$ becomes less than $I_{full}$ (for the ideal contribution of $I_{vertical}$ and $I_{in-plane}$ see Supplementary Fig. 3). All of these observations prove that the vertical asymmetry is indeed able to generate a net CPGE current by changing the distribution of the carriers in the band structure, although they do not exclude the possibility of local in-plane asymmetries, such as contacts or edges.

**Simulation of the band structure and the selection mechanism.** In order to further investigate the origin of CPGE in PbS nanosheets, we calculated the band structure of clean PbS (001) slabs (with 15 atomic layers) including an external electric field using DFT based on the general-gradient approximation, employing the Perdew–Burke–Ernzerhof (PBE) exchange-correlation functional[28] (see Supplementary Fig. 4). SOC was taken into account by using fully relativistic projector-augmented-wave (PAW) potentials. It should be pointed out that when SOC is considered, general-gradient approximation functionals are known to underestimate the band gap in bulk PbS and even predict a band inversion[29,30]. The usage of hybrid functionals (for example, Heyd–Scuseria–Ernzerhof) or many-body corrections like the GW approximation were shown to improve the agreement with experimental band gaps, but they are much more computationally demanding than pure DFT[29,30]. The qualitative trend of the influence of an external electric field on the band structure is expected to be described well enough by PBE to get reliable results, as this functional was already used to describe Rashba systems[31–33].

Without an external electric field, inversion symmetry is maintained, and no Rashba spin splitting can be observed along

the Γ–M–X path (see Supplementary Fig. 5a). At the band gap (M point), the two highest filled bands split in direction of the Γ point. Adding an external electric field along the surface normal breaks the inversion symmetry, and Rashba spin splitting occurs at the M point (see Supplementary Fig. 5b), which is confirmed by a spin texture typical for the Rashba effect (see Supplementary Fig. 5c). The Rashba spin splitting of the highest filled band differs from the splitting of the second highest filled band. Rashba spin splitting can be also observed for the conduction band.

The calculated band structure implies that by exciting the material with right-handed (left-handed) polarized light, that is photons with angular momentum of +1 (−1), some of the possible transitions between the valence and conduction bands are forbidden since energy and angular momentum must be conserved during the transition. The possible excitation mechanism of the carriers over the band gap is illustrated schematically in Fig. 2b. The bands are split around the M point. However, in contrast to other multi-valley materials, the splitting occurs in momentum space and the split bands, corresponding to the two opposite spin orientations, have equal energies. For materials with rectangular Brillouin zone, both valleys (M and M′) represent similar orbital characters. Therefore, they are not selectively populated by adjusting the angular momentum of the exciting photons (which is done by selecting the handedness of the circular polarization)[27]. On the other hand, by splitting the bands in momentum space, the angular momentum of the split bands becomes different for each spin. As it can be observed in Supplementary Fig. 6, total angular momentum of the valence band and the conduction band are respectively 3/2 and 1/2 with spin-angular momentum of 1/2 (ref. 34). By controlling the helicity of the exciting light, carriers can be selectively excited based on their spin orientation[12,14,26]. Upon illumination with circularly polarized light and population of the conduction band with spin-polarized electrons, the number of excited carriers is equal in the M valley and in the M′ valley, but the linear momentum differs, since splitting is asymmetric at these two points. For instance, the spin-up band is shifted away from the Gamma point (to higher momentum) at the M point, but towards the Gamma point (to lower momentum) at M′. Therefore, spin-up electrons at the M point have higher momentum compared to those at the M′ point. This results in the generation of the spin-polarized CPGE current based on Rashba SOC which splits the valleys of the band structure.

The current generated by the CPGE in the direction of a specific crystal orientation (λ) can be expressed by

$$J_{\text{CPGE}-\lambda} = \sum_{\mu} \chi_{\lambda\mu} e_{\mu} E_0^2 P_{\text{circ}}, \qquad (2)$$

where μ is one of the different crystal orientations $(x, y, z)$, χ is the CPGE second-rank pseudo tensor which is directly affected by the crystal asymmetry, $\mathbf{E}_0$ is the electric field (complex amplitude) of the light wave, $P_{\text{circ}}$ is the helicity of the circularly polarized light (degree of polarization), $\mathbf{e} = \mathbf{q}/q$ is the unit vector for the light propagation, and $\mathbf{q}$ is the wave vector of the light in the medium[5,6,9]. For the $C_{4v}$ point group, the pseudo tensor χ has non-zero elements, which results in a non-zero CPGE current[35,36].

**Gate dependency of the effect.** Having introduced confinement and symmetry breaking by the gate electric field as the factors influencing the SOC, we separately studied these elements under the low illumination angle in order to tune the band splitting in the PbS nanosheets. Figure 3 shows the dependence of the CPGE current on the gate voltage (see also Supplementary Fig. 7). Changing the gate voltage can influence the photocurrent in

two ways: First, it affects the band splitting by modification of the inversion asymmetry[5,6,12]. Second, the band alignment of the material with the contact metals is altered, which results in changing the extraction probability and the recombination rate of all the excited carriers including the spin-polarized ones[37–39]. The former is observable only for the CPGE current, whereas the latter is a general effect and can govern all three components of the photocurrent ($J_0$, $J_{\text{CPGE}}$, $J_{\text{LPGE}}$). As a result, comparable responses to the gate can be observed for these components (Fig. 3a). In order to extract the pure gate dependence of SOC, the change in the band alignment or any other spin-independent effect should be excluded from the results. This can be done by normalizing the CPGE current by the background current[6,9,16]. The normalized value represents the internal tunability of the band splitting by the gate voltage.

As shown in Fig. 3b, by sweeping the back-gate voltage from −8 to 8 V, the CPGE current of the nanosheets changes significantly. The gate dependency of the normalized CPGE current also shows the clear tunability of the band splitting by the gate electric field. By changing the electric field in the

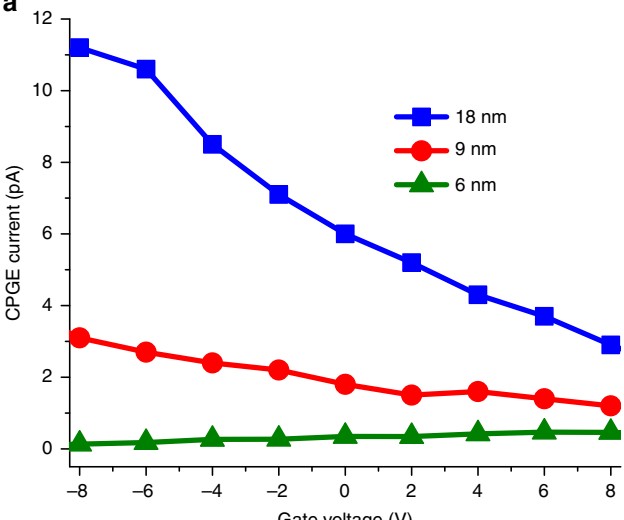

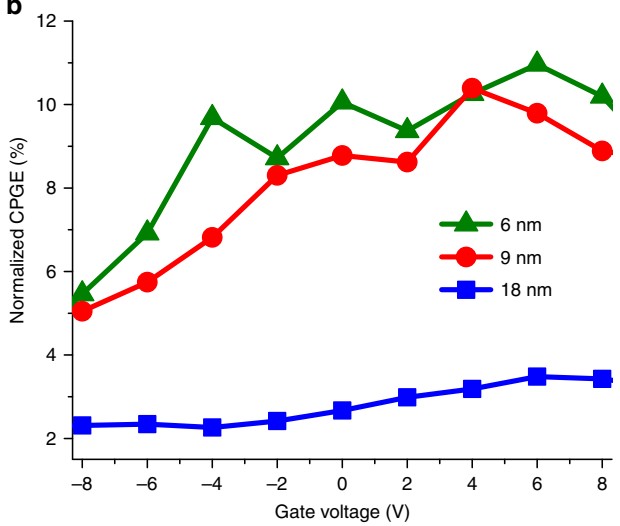

**Figure 4 | Circular photo-galvanic current for different thicknesses.** (**a**) The CPGE current for 6, 9 and 18 nm thick sheets, measured as a function of the gate voltage. (**b**) CPGE current normalized by the background current, as a function of the gate voltage. Thinner sheets show higher gate dependencies.

semiconductor channel, resulting from the back gate, the potential profile of the confined crystal can be modulated[6], resulting in the modulation of the degree of reduction in symmetry. As a result, by increasing the gate voltage, a larger band splitting can be expected[12], which causes a wider distribution of the carriers in momentum space and hence a higher CPGE current is generated. The DFT band structure calculations show that the strength of the gate electric field can modify the Rashba splitting of the valence and conduction bands (Fig. 3c). The gate effect is superimposed on the boundary conditions resulting from the asymmetric interfaces. Therefore, even at zero gate voltage, a CPGE current is generated (Fig. 3b). Dependency on the gate voltage is another confirmation that the vertical asymmetry is effective in the CPGE, since the vertical electric field is not expected to affect the in-plane asymmetry.

**Thickness dependency of the effect**. Moreover, we performed the CPGE measurements on devices based on nanosheets with three different thicknesses of 6, 9, and 18 nm to investigate the influence of the quantum confinement on the Rashba effect (Fig. 4a and Supplementary Fig. 8). By sweeping the gate voltage, it can be seen that thicker sheets produce higher CPGE currents. The character of the gate dependency for the CPGE current is similar to the background current for all devices. Since the channel dimensions (thickness and width) are different for each sample, the absorption capability of the devices is different as well. In order to make the results comparable, the CPGE normalized by the background current was calculated (Fig. 4b). The measurements indicate that by decreasing the thickness of the sheets, which corresponds to an increase of the confinement effect, structural inversion asymmetry and consequently the Rashba SOC becomes more pronounced in PbS nanosheets. Despite the different gate dependencies of the CPGE currents, all thicknesses of nanosheets show comparable gateabilities for the normalized CPGE. To clearly see the effect of the thickness on the vertical asymmetry, tunability of the normalized CPGE with the gate voltage can be observed, which is only a character of the band splitting due to the vertical asymmetry and excludes the effects of any in-plane asymmetry. As shown in Fig. 4b, the thinner the nanosheet, the more effective is the gate voltage. By sweeping the back-gate voltage from − 8 to 8 V, the normalized CPGE for the 6 nm sheets changes six times more than that for the 18 nm sheets. The gate voltage and the asymmetric boundaries result in an electric field in the crystal which breaks the symmetry. The thinner the nanosheets the stronger is the effective field. By increasing the effective electric field in the crystal, splitting becomes larger, and a stronger CPGE can be detected.

## Discussion

A CPGE has been observed in colloidal PbS nanosheets, which could be assigned to Rashba SOC. The inversion symmetry in the rock salt crystal of bulk PbS was broken by quantum confinement, by asymmetric interfaces on two sides of the material, and by a gate electric field. The effect of these parameters was investigated experimentally, and the results were substantiated by DFT simulations of the band structure. The latter shows a splitting of the bands in momentum space, which results in an unconventional selection mechanism based on the spin of the photo-excited carriers. Our results are consistent with a higher Rashba SOC in thinner sheets. The observation of spin-related electrical transport phenomena in colloidal materials is promising in terms of future industrial applications, which supports the recently emerging spintronic approaches with simplicity, inexpensiveness and scalability of the colloidal synthesis of nanomaterials.

## Methods

**Synthesis**. All chemicals were used as received: lead(II) acetate tri-hydrate (Aldrich, 99.999%), thioacetamide (Sigma-Aldrich, ⩾99.0%), diphenyl ether (Aldrich, 99% + ), dimethyl formamide (Sigma-Aldrich, 99.8% anhydrous), oleic acid (Aldrich, 90%), trioctylphosphine (ABCR, 97%), 1,1,2-trichloroethane (Aldrich, 96%). In a typical synthesis, a three neck 50 ml flask was used with a condenser, septum and thermocouple. 806 mg of lead acetate trihydrate (2.3 mmol) was dissolved in 10 ml of diphenyl ether. Depending on the targeted thickness, 2–10 ml of OA (5.7–28 mmol) was added. The mixture was heated to 75 °C until the solution turned clear. Then, vacuum was applied for 3.5 h to transform lead acetate into lead oleate and to remove acetic acid in the same step. The solution was heated under nitrogen flow to the desired reaction temperature of 130 °C, while at 100 °C, 0.7 ml of TCE (7.5 mmol) was added under reflux to the solution and the time has been started. After 12 min 0.23 ml of 0.04 g TAA (0.5 mmol) in 6.5 ml DMF was added to the reaction solution. After 5 min, the heat source was removed and the solution was let to cool down below 60 °C which took approximately 30 min and afterwards, centrifuged at 4,000 rpm for 3 min. The precipitant was washed two times in toluene before the nanosheets were finally suspended in toluene again for storage.

**Device preparation**. PbS nanosheets with lateral dimensions of up to 5 μm suspended in toluene were spin-coated on silicon wafers with 300 nm thermal silicon oxide as the gate dielectric. The highly doped silicon was used as the back gate. The individual nanosheets were contacted by e-beam lithography followed by thermal evaporation of Ti/Au (1/40 nm) and lift-off.

**Measurements**. Immediately after device fabrication, we transferred the samples to a probe station (Lakeshore-Desert) connected to a semiconductor parameter analyser (Agilent B1500a). All the measurements have been performed in vacuum at room temperature. The vacuum chamber had a view port above the sample which is used for sample illumination. For illumination of the nanosheets, an 18 mW red laser (627 nm) with a spot size of 4 mm was used. The laser was able to excite the electrons over the band gap, while its energy was not enough for excitations to higher levels of the conduction band[40]. The polarization of the laser beam was controlled by a polarization filter and a quarter-wave plate.

**Density functional theory simulations**. Density functional theory calculations were done by employing the electronic structure code Quantum Espresso 5.2.1 (ref. 41) in combination with the PBE exchange-correlation functional[28] and the PAW method. Relativistic effects (scalar- and spin–orbit coupling) were considered through the PAW potentials (Valence configuration: Pb: $5d^{10}6s^26p^2$; S: $3s^23p^4$) self-consistently (the used PAW potentials for lead and sulphur were taken from the pslibrary.1.0.0 downloadable at: http://www.qe-forge.org/gf/project/pslibrary/frs). A kinetic-energy cut-off for the wavefunction of 46 Ry and a kinetic-energy cut-off of the electronic density of 460 Ry were used. The Brillouin zone was sampled with a shifted $8 \times 8 \times 8$ grid for bulk PbS and a shifted $8 \times 8 \times 1$ mesh for two-dimensional (001)-PbS sheets. The default convergence thresholds were used for all calculations.

The atomic positions in the slab were allowed to relax, while the lattice constant was kept fixed at the value calculated for bulk PbS using the same computational settings (6.002 Å). SOC and dispersion interactions were not considered during the relaxation. The slabs were separated by a vacuum of 15 Å.

Band-structure calculations were carried out on those optimized structures considering SOC. External electric fields were simulated with a saw-like potential, changing along the surface normal. A larger vacuum layer (20 Å) was used, and the decrease of the saw-like potential to its initial value was set to be in the middle of the vacuum. Symmetry was not used to reduce the number of **k** points during the self-consistent field calculation.

The projected density of states resolved on the band structure was calculated using a Gaussian smearing with a broadening of 0.0001 Ry. The projected density of states was summed over all atoms for any type of orbital.

**Data availability**. All the theoretical and experimental data supporting this study are available from the corresponding author.

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

## Acknowledgements

M.M.R.M., T.B. and C.K. gratefully acknowledge financial support of the European Research Council via the ERC Starting Grant '2D-SYNETRA' (Seventh Framework Program FP7, Project: 304980). C.K. thanks the German Research Foundation DFG for financial support in the frame of the Cluster of Excellence 'Center of ultrafast imaging CUI' and the Heisenberg scholarship KL 1453/9-2. M.S.Z. and C.H. thank the DFG for funding via the project SFB668 (project B17) and the North-German Supercomputing Alliance (HLRN) for computational resources. The authors thank Vladimiro Mujica (Arizona State University) for discussion and Sascha Kull for support in the synthesis of the nanosheets.

## Author contributions

M.M.R.M. and C.K. conceived the main concepts. M.M.R.M. designed the experiments, prepared the samples and performed the electrical measurements. T.B. performed the nanosheets synthesis and characterization. M.S.Z. and C.H. performed the simulations and theoretical analysis of the results. M.M.R.M., T.B., M.S.Z., C.H. and C.K. wrote the manuscript.

## Additional information

**Competing interests:** The authors declare no competing financial interests.

