## [Peer Review File · Nature Communications]

Reviewers' Comments:

Reviewer #1 (Remarks to the Author):

The CPGE was observed in colloidal PbS nanosheets, which is caused by Rashba SOC. It was illustrated the SOC can be influenced by the symmetry and tuned by the back gate. The study is interesting. Before the publication of the work, some questions as follows should be answered.

1. The photon energy of the excited laser is much larger than the band gap at $M(M')$ point. So the electrons in the valence band should be excited to the higher band (not the conduction band minimum), which is not clear in the calculations. Thus, long wavelength laser should be used in the experiments.

2. The schematic illustration of the geometry is not clear. What is the incident plane? The accurate angle of the high incident angle and low incident angle (Line 130, Fig.1a, Fig.2a)?

3. The illustration of Fig. 2b is puzzling. The angular momentum conservation is not clear. What are the orbital and spin angular momentum of the band? Additionally, the group velocity of the excited electrons at M point (the position of the two arrows) should have opposite sign. And it should be explained that why the velocity at K_m is not equal to that at K_m' . I cannot see it according to the symmetry of the split band.

4. The physical mechanism of the I-vertical and I-inplane is not clear. It needs a schematic illustration like Fig.2b combined with the symmetry analysis of pseudo tensor X (Line 200). For example, there is no CPGE along c -direction excitation in wurtzite structure.

Reviewer #2 (Remarks to the Author):

The authors demonstrate the observation of Rashba-splitting in nanosheets of PbS. Thin layers of PbS of a few nm thickness and a few micrometers in lateral dimension were synthesized by a relatively simple solution based method, described by some of the authors in previous papers. This method allows obviously a good control of the thickness of the sheets quite well. The spin coated samples (devices) were then investigated. The observed circular photo-galvanic effect (CPGE) could be explained by the Rashba SOC. The results are supported by an interpretation results from KS-DFT calculations.

The results are of high current interest, especially because the authors could show the Rashba effect on colloidal nanoparticles that can be synthesized in an easy but controllable way instead of elaborate other techniques.

The materials and methods are well described – see below.

The KS-DFT calculations - considering relativistic effects - are appropriate for such study. Also the application of the PBE exchange-correlation functional is well justified for the description of the qualitative trend and the consideration of the influence of the external electric field. The calculations seem to be done properly.

As a minor point: It is not clear to me what the authors mean with “SOC was taken into account self consistently”, especially in connection with the statement in the methods section: “Relativistic effects were considered through the PAW potentials”. The authors should clarify this issue.

Altogether, the paper is worth for publication in Nature Communications after minor revision – see above.

Answers to the reviewers

(Our answers in italic)

We are very happy that the Referees like our work. We thank them for the time they took to read our manuscript and for the useful suggestions. In the following we answer to their specific comments.

Reviewer 1

Reviewer: 1. The photon energy of the excited laser is much larger than the band gap at M(M') point. So the electrons in the valence band should be excited to the higher band (not the conduction band minimum), which is not clear in the calculations. Thus, long wavelength laser should be used in the experiments.

Answer: *As it is mentioned in the text, using GGA functionals normally leads to an underestimation of the band gap. Approaches giving larger band gaps, such as hybrid functionals or DFT+U, are considerably more costly computationally and beyond the scope of this work (where our focus is on qualitative considerations). By considering the realistic values of the band structure for bulk PbS (see following figure), it can be concluded that the energy of the used laser is not enough to excite the electrons to higher bands. Since the nanosheets are strongly confined, the required energy for such excitation is even higher than the provided values, which emphasizes the improbability of excitation to higher bands.*

*Realistic band structure of bulk PbS (F. Herman, R. Kortum, I. Ortenburger, J. P. Van Dyke, Relativistic band structure of GeTe, SnTe, PbTe, PbSe, and PbS, J. Phys. Colloques **29**, C4-62-C4-77, 1968).*

Changes: To make this clearer, we added the following sentence to the Materials and Methods: “The laser was able to excite the electrons over the band gap, while its energy was not enough for excitations to higher levels of the conduction band⁴⁰.” We also added Ref. 40 to illustrate the realistic values of the band structure.

Reviewer: 2. The schematic illustration of the geometry is not clear. What is the incident plane? The accurate angle of the high incident angle and low incident angle (Line 130, Fig.1a, Fig.2a)?

Answer: We would like to thank the referee for this hint. As it can now be seen in the modified version of Fig. 1a, the incidence plane is perpendicular to the direction of the current flow. Concerning the incidence angles, the values were added to the text. Figure 2a (inset) is the top view of the setup shown schematically in Fig. 1a.

Changes: We modified Fig. 1a in order to make the schematic view clearer. We added the following sentence to the text: “The beam was pointed to the sample obliquely in the yz plane (x is the direction of the current flow, z is normal to the nanosheet, and y is perpendicular to these two).” Further, we expanded the explanation about the incidence angles to: “The measurements were done with a high incidence angle (about 15 degrees, the max. incidence angle due to the size of the view port of the vacuum chamber) and a lower one (around 6 degrees).” Eventually, we updated the following sentence in the caption of Fig. 2: “The inset is a schematic top view of the device to illustrate the position of the shadow during the experiment.”

Reviewer: 3. The illustration of Fig. 2b is puzzling. The angular momentum conservation is not clear. What are the orbital and spin angular momentum of the band? Additionally, the group velocity of the excited electrons at M point (the position of the two arrows) should have opposite sign. And it should be explained that why the velocity at Km is not equal to that at Km'. I cannot see it according to the symmetry of the split band.

Answer: We carried out additional calculations to elucidate the nature of the bands. They suggest that the largest contribution to the valence band comes from the $p_{3/2}$ orbital of sulfur (orbital angular momentum $L=1$, spin-angular momentum $S=1/2$, total angular momentum $J=|L+S|$), and that the largest contribution to the conduction band comes from the $p_{1/2}$ orbital of lead ($L=1$, $S=1/2$, $J=|L-S|$). Possible transitions which conserve angular momentum, therefore, are (where states are described as $|J, J_z\rangle$):

1. $|3/2, +3/2\rangle \rightarrow |1/2, +1/2\rangle$
2. $|3/2, -3/2\rangle \rightarrow |1/2, -1/2\rangle$
3. $|3/2, +1/2\rangle \rightarrow |1/2, -1/2\rangle$
4. $|3/2, -1/2\rangle \rightarrow |1/2, +1/2\rangle$

Among the mentioned possibilities, transitions from $|3/2, \pm 3/2\rangle$ are more probable than other ones²⁶. Thus, the selection mechanism of our experiment which conserves the angular momentum and leads to an asymmetric distribution of carriers in momentum space can be:

1. $|3/2,+3/2\rangle \rightarrow |1/2,+1/2\rangle$
2. $|3/2,-3/2\rangle \rightarrow |1/2,-1/2\rangle$

Based on this conclusion, we modified Fig. 2b. We also added the character of each band to explicitly illustrate the conservation of the angular momentum.

The group velocity at the M point has an opposite sign in comparison to the M' point (since they represent opposite directions in the lattice). The magnitude of the velocity is also unequal at the M and the M' point, as splitting is asymmetric at these two points. For instance, the spin-up band is shifted away from the Gamma point (to higher momentum) at the M point, but towards the Gamma point (to lower momentum) at M'. Therefore, spin-up electrons at the M point have higher momentum compared to those at the M' point. By populating the conduction band at the M and the M' point with spin-polarized electrons, distribution of the carriers in momentum space would be asymmetric, leading to the generation of a net current.

Changes: We modified Fig. 2b. The following sentence was added to the caption of Fig. 2b: "Here, the angular momentum of exciting photons is -1, which is added to the spin-angular momentum of electrons." This sentence was added to the main text too: "As it can be observed in Fig. S6 in the Supplementary Information, total angular momentum of the valence band and the conduction band are respectively 3/2 and 1/2 with spin-angular momentum of 1/2³⁴." The last part of the explanation about the selection mechanism was updated to: "Upon illumination with circularly-polarized light and population of the conduction band with spin-polarized electrons, the number of excited carriers is equal in the M valley and in the M' valley, but the linear momentum differs, since splitting is asymmetric at these two points. For instance, the spin-up band is shifted away from the Gamma point (to higher momentum) at the M point, but towards the Gamma point (to lower momentum) at M'. Therefore, spin-up electrons at the M point have higher momentum compared to those at the M' point. This results in the generation of the spin-polarized CPGE current based on Rashba spin-orbit coupling which splits the valleys of the band structure." Reference 34 was added. Finally, we added a picture of projected-density-of-states-resolved band structures in the vicinity of the band gap to the Supplementary Information. This shows the character of the bands with respect to the total angular momentum and therefore provides information on possible excitation mechanisms, which follow the conservation of momentum. The required procedure for achieving this picture was explained in the Methods: "The projected density of states (PDOS) resolved on the band structure was calculated using a Gaussian smearing with a broadening of 0.0001 Ry. The PDOS was summed over all atoms for any type of orbital."

Reviewer: 4. The physical mechanism of the I-vertical and I-inplane is not clear. It needs a schematic illustration like Fig.2b combined with the symmetry analysis of pseudo tensor X (Line 200). For example, there is no CPGE along c-direction excitation in wurtzite structure.

Answer: In order to clarify the achieved results in Fig. 2, the current produced by Rashba SOC has been labeled in this part as I-vertical, since vertical asymmetry is responsible for its generation (its physical mechanism is explained comprehensively in the text). I-inplane is the generated current due to any in-plane asymmetries in the structures (e.g. contacts, edges or position of the beam). Although

this part of the current is dependent on the light helicity, (normally) it is not spin polarized. In the text, we do not suggest any possible mechanism for I-inplane (some of the possible mechanisms can be found in Ref. 12, 16, 25) since it does not originate from Rashba SOC, which is the main focus of this work. Instead, as undesirable side effects which cannot be avoided, they were considered in the evaluations separately from the main phenomenon (Rashba SOC) to prove the origin of the observed CPGE.

Reviewer 2

Reviewer: As a minor point: It is not clear to me what the authors mean with “SOC was taken into account self consistently”, especially in connection with the statement in the methods section: “Relativistic effects were considered through the PAW potentials”. The authors should clarify this issue.

Answer: *The spin-orbit coupling was considered by using fully-relativistic PAW potentials. By “SOC was taken into account self consistently”, we wanted to point out that spin-orbit coupling was not described with perturbation theory, but was taken into account (via the PAW potentials) during the self-consistent field algorithm for solving the Kohn-Sham equations.*

Changes: *To remove any source of confusion we replaced this sentence by “SOC was taken into account by using fully-relativistic PAW potentials”.*

Reviewers' Comments:

Reviewer #1 (Remarks to the Author):

The answers of my questions is satisfied. It can be published now.

Reviewer #2 (Remarks to the Author):

The authors have thoroughly answered my question and properly considered in the revised version of the manuscript.

Therefore, I can now recommend the publication of the paper in Nature Communication in its present form.